# Unveiling the Molecular Landscape of Pancreatic Ductal Adenocarcinoma: Insights into the Role of the COMPASS-like Complex

**DOI:** 10.3390/ijms25105069

**Published:** 2024-05-07

**Authors:** Marzieh Jamali, Erfaneh Barar, Jiaqi Shi

**Affiliations:** 1Department of Pathology & Clinical Labs, Rogel Cancer Center, University of Michigan, Ann Arbor, MI 48109, USA; 2Liver and Pancreatobiliary Diseases Research Center, Digestive Disease Research Institute, Shariati Hospital, Tehran University of Medical Sciences, Tehran 1416634793, Iran

**Keywords:** PDAC, *KMT2D*, *KMT2C*, *KDM6A*, *KMT2*

## Abstract

Pancreatic ductal adenocarcinoma (PDAC) is poised to become the second leading cause of cancer-related death by 2030, necessitating innovative therapeutic strategies. Genetic and epigenetic alterations, including those involving the COMPASS-like complex genes, have emerged as critical drivers of PDAC progression. This review explores the genetic and epigenetic landscape of PDAC, focusing on the role of the COMPASS-like complex in regulating chromatin accessibility and gene expression. Specifically, we delve into the functions of key components such as *KDM6A*, *KMT2D*, *KMT2C*, *KMT2A*, and *KMT2B*, highlighting their significance as potential therapeutic targets. Furthermore, we discuss the implications of these findings for developing novel treatment modalities for PDAC.

## 1. Introduction

It is anticipated that pancreatic cancer, specifically pancreatic ductal adenocarcinoma (PDAC), will be ranked as the second leading cause of cancer-related death by 2030 [1,2,3]. The survival rate for PDAC has improved substantially over the past decades but remains low at 12% [4]. Although surgical resection offers the only curative option for patients without metastasis, the majority of those who undergo a curative pancreatectomy will experience recurrence post-surgery. Additionally, PDAC presents with aggressive and often asymptomatic features, leading to the diagnosis of most patients at a metastatic stage, where fewer than 20% are eligible for surgical resection [5,6,7]. Therefore, most individuals diagnosed with PDAC need to receive a combination of chemotherapy; however, this results in a median overall survival period of less than one year [5,6,8,9]. Given the inadequacy of current therapeutic approaches, it is evident that innovative and more effective strategies are urgently needed. Aligned with this aim, genetic and epigenetic alterations are crucial in driving cancer cell progression. Understanding the mechanisms underlying these alterations can help identify potential targets for treating PDAC [10,11,12,13]. Genetic studies utilizing whole-exome and gene expression analyses have identified mutations in four tumor driver genes—*KRAS*, *TP53*, *SMAD4*, and *CDKN2A*—as molecular fingerprints for PDAC, with KRAS being prevalent in nearly all cases of the disease [14,15,16,17,18,19]. While these key driver mutations influence prognosis, diagnosis, and resistance to chemotherapy, targeting these driver mutations has been challenging [20,21,22]. However, recent promising preclinical studies using a KRAS^G12D^ inhibitor in PDAC shined new light on targeting these “untargetable” mutations and led to a phase I/II clinical trial [23,24].

In addition to the above-mentioned four driver mutations, up to a quarter of PDACs, other solid tumors, and hematological malignancies exhibit gene alterations linked to the COMPASS (complex of proteins associated with SET1)-like complex [25,26,27,28,29]. An extensive study across various cancer types classified the COMPASS-like complex as the third most commonly mutated complex in squamous-like cancers, which was associated with a poor prognosis [30]. It has been demonstrated that genes belonging to the COMPASS-like complex can function as oncogenic modulators in medulloblastoma, non-small cell lung cancer, and leukemia, and depletion of components within this intricate complex leads to significant alterations in promoter activity and aberrant gene transcription [26,29,31,32]. It is conceivable that pro-tumor pathways could contribute to the depletion of components within the COMPASS-like complex. This disruption may interfere with transcription factors essential for tumor suppression and anti-tumor immune responses [33,34]. This suggests that COMPASS-like complexes may represent a burgeoning avenue in cancer research, particularly in cancers characterized by poor differentiation, squamous features, and aggressive behavior [35].

Recent studies show alterations in COMPASS-like genes have been associated with PDAC, especially those with an aggressive nature, poor differentiation, and squamous-like differentiation [35,36]. Therefore, the COMPASS-like complex emerges as a promising therapeutic target for PDAC. Consequently, this review examines PDACs exhibiting genetic alterations within the COMPASS-like complex and aims to deepen our understanding of the clinicopathologic and genetic characteristics of this molecularly distinct subgroup within pancreatic cancer. Additionally, we explore how this molecular insight can be translated into practical applications, potentially offering new avenues for identifying patients with more aggressive diseases and unlocking therapeutic potential.

## 2. COMPASS-like Complex Gene Interactions in Cancer

A growing body of research is currently exploring the role of COMPASS-like complex genes in the epigenetic regulation of cancer. This complex is also under investigation concerning PDAC, considered the most lethal solid tumor. Key components of the COMPASS-like complex, including *KDM6A* and the *KMT2* family (encompassing *KMT2A*, *KMT2B, KMT2C*, and *KMT2D*), play crucial roles in modulating chromatin accessibility in PDAC through epigenetic mechanisms [13,35,37]. Histone methylation, a critical mechanism of epigenetic regulation, is influenced by enzymes encoded by genes within the *KDM6A* and *KMT2* family. These enzymes, acting as methyltransferases or demethylases, modify histone proteins by adding or removing methyl groups at specific lysine residues (Table 1) [32,38,39,40,41,42,43].

KMT enzymes catalyze various lysine methylation reactions on core histones using S-adenosyl-L-methionine as a methyl donor [44,45]. Alternatively, the Jumonji-C (JmjC) domain in KDM enzymes facilitates demethylation by oxidizing the methyl group, indicating the importance of JmjC in regulating gene expression and chromatin structure [46,47]. The functions and interactions within both the KMT and KDM families are governed by plant homeodomain fingers (PHD Fingers), enabling proteins to recognize and interact with chromatin and consequently playing critical roles in regulating gene expression, chromatin structure, and other cellular activities. The precise significance of PHD fingers and their potential relevance in pancreatic cancer remains to be fully elucidated [48,49,50]. Consequently, COMPASS-like genes impact the recruitment of transcriptional machinery, chromatin remodeling, and gene expression, ultimately influencing the fate of pancreatic cancer cells.

*KDM6A* (lysine (K)-specific demethylase 6A), situated on the X chromosome and also referred to as *UTX* (ubiquitously transcribed X chromosome tetratricopeptide repeat protein), is recognized as one of the most significant epigenetic factors for PDAC [38]. A primary role of KDM6A is to govern gene expression and chromatin accessibility by catalyzing the demethylation of histone 3 lysine 27 (H3K27) [51]. As a result, KDM6A is implicated in the regulation of various cellular processes, including cell growth, apoptosis, and differentiation. This suggests the potential involvement of KDM6A in driving cancer progression [52,53]. Regarding PHD fingers within the KDM family, it is worth noting that although the PHD1 finger’s location near the JmjC domain in KDMs such as KDM5C assists in substrate recognition, stabilization, and H3K4 positioning, it is not indispensable for demethylation [54,55,56]. Conversely, PHD fingers in KDMs such as KDM5A and KDM5B play a role in substrate binding, influencing histone demethylation activity by preferentially recognizing specific histone marks like unmodified histone H3 (H3K4me) and H3K4me3 [50,57,58].

While KDM6A has been identified to exert tumor suppressive effects in several cancers, such as lung, hematologic, bladder, and breast cancer, it is noteworthy that KDM6A exhibits tumor-promoting activity in cervical cancer [59,60,61,62,63,64,65,66,67]. It has been demonstrated that KDM6A is frequently inactivated in PDAC, suggesting that it may play a tumor-suppressive role [36,68]. According to further studies, KDM6A contributes to differentiation, normal growth, cell migration and invasion, and tumor suppression independent of its demethylase activity in pancreatic cancer cells [53,69]. As KDM6A plays an essential role in the differentiation, migration, and growth regulation of pancreatic cancer cells, deletion of this gene results in squamous differentiation, invasion, metastasis, and poor prognosis of PDAC [36,69]. KDM6A is also capable of serving as an independent prognostic factor for PDAC [8,36,70]. Therefore, KDM6A has the potential to serve as a diagnostic, prognostic, and therapeutic target for PDAC. PDAC is also affected by other members of the KDM enzyme family. While KDM6A and KDM6B have a tumor suppressor role, other members of the KDM family like KDM2B, KDM3A, and KDM5A may serve as oncogenes in PDAC [58,71,72]. Overexpression of *KDM2B* and *KDM5A* genes plays significant roles in promoting poor differentiation and tumor proliferation in PDAC, respectively, whereas knockdown of the *KDM3A* gene leads to decreased invasive activities in culture and impedes the formation of orthotopic tumors in mice [58,71,73]. The knockdown of KDM6B in PDAC cell lines accelerates tumor growth regardless of KRAS status, despite KDM6B being a downstream target of KRAS [58,72]. As a result of these findings, the KDM enzyme family plays a range of functions in PDAC and may serve as potential therapeutic targets.

Similar to mutations in *KDM6A*, members of the type 2 lysine methyltransferase (*KMT2*) family (also known as *MLL*) are mutated in various types of cancer, suggesting that they play an important role in tumorigenesis [32]. The KMT2 family consists of the histone lysine methyltransferases KMT2A (MLL1), KMT2B (MLL2/MLL4), KMT2C (MLL3), and KMT2D (MLL2/MLL4) that act as catalytic subunits within mammalian COMPASS-like complexes to add methyl groups to histone H3 (H3K4) [41]. While the KMT2A/KMT2B complexes are involved in di- and tri-methylating H3K4 (H3K4me2/3), the KMT2C and KMT2D complexes are involved in monomethylating H3K4 (H3K4me1) [42,43,74,75,76]. In contrast to KMT2A and KMT2B, KMT2C and KMT2D inhibit cell proliferation and are often considered tumor suppressors in different types of cancer [32,77,78,79,80,81]. *KMT2* family member mutations are an independent prognostic factor associated with PDAC survival [43,81,82]. KMT2 proteins all possess a SET domain, termed after three proteins found in Drosophila (including su(var)3-9, enhancer-of-zeste, and trithorax), which catalyzes the transfer of methyl groups toward lysine residues. Drosophila components, including trithorax (Trx) and trithorax-related (Trr), are also involved in the cluster of KMT2 complexes [44,45,83,84]. The Trx-related subgroup contains KMT2A and KMT2B, while the Trr-related subgroup contains KMT2C and KMT2D. Both the KMT2A/2B complex and KMT2C/2D complex are associated with promoter and enhancer regions and share essential components such as WDR5, ASH2, hDPY30, and RBBP5. Specific components belonging to the KMT2A/2B complexes are Menin and HCF1/2, whereas the KMT2C/2D complex includes specific subunits like the histone lysine demethylase PTIP (PAX transactivation domain-interacting protein), PA1 (PTIP-associated 1), NCOA6 (nuclear receptor coactivator 6), and KDM6A [29,84,85,86,87,88]. Within KMT2C/2D complexes, KDM6A removes the methyl group from the marked histone, enabling the acetylation of H3K27, and thereby facilitating the activation of the enhancer [89,90]. PHD fingers are crucial in regulating KMTs such as KMT2A, KMT2B, and KMT2C that engage intramolecular interactions, holocomplex formation, and protein structure stabilization. The presence of PHD1-3 fingers in KMT2A and KMT2B can modulate protein stability and interactions with histone marks like H3K4me3 [48,91,92,93]. PHD4-6 in KMT2D play a crucial role in nucleosomal methylation activity and stem cell differentiation [58,94].

Among the members of the *KMT2* family, *KMT2C* and *KMT2D* are frequently mutated in several types of human cancer [29,77,95]. KMT2D is essential for forming active promoter and enhancer landscapes and constructing transcriptomes unique to individual cells [96,97]. The beta-globin and estrogen receptor genes are transcriptionally regulated by KMT2D [98]. In addition to the role that KMT2D plays in developmental disorders like Kabuki syndrome [99], it has been found to play a dual role in various types of cancer [100]. Although KMT2D can act as a tumor suppressor in some cancers, such as medulloblastoma or bladder cancer, in esophageal squamous cell carcinoma cells, KMT2D knockout inhibited cell proliferation and migration and reduced epithelial-mesenchymal transition [26,101,102]. It has been shown that the expression level of KMT2D may change the prognosis of some cancers, including non-small cell lung cancer and esophageal squamous cell carcinoma [90,103]. In PDACs, *KMT2D* is frequently transcriptionally repressed through DNA methylation, leading to decreased expression, which correlates with increased proliferation, invasion, and migration [104,105,106,107]. Through modulation of glucose and lipid metabolism and cell plasticity, low expression levels of KMT2D cause proliferation, the epithelial–mesenchymal transition, and tumor progression in PDAC [104,108]. These suggest the tumor-suppressive role of KMT2D, supported by its association with a gene signature resembling the aggressive Moffitt basal-like subtype and indicating a poorer prognosis compared to the classical subtype [106,107,108]. In more than 80% of neoadjuvant-treated PDACs, *KMT2D* mutations were detected, indicating that the mutation may be responsible for treatment resistance [82]. KMT2D’s role in chromatin remodeling and gene expression regulation makes it a valuable target in PDAC research and therapy.

Another *KMT2* family member frequently mutated in cancer is *KMT2C*, which regulates enhancer activity in normal cells through methylation [33,74,77]. Nevertheless, recent research suggests that KMT2C may also be involved in transcriptional regulation independent of its methylation ability [34,109,110]. In urothelial carcinoma, colorectal cancer, and acute myeloid leukemia, KMT2C has been shown to function as a tumor suppressor, although its role in tumorigenesis remains largely uncertain [32,80,110,111,112]. The combined action of KMT2D and KMT2C results in the formation of a complex through which KDM6A exerts its enhancer-regulating function [76,113,114]. By acting as an essential subunit of the KMT2C/D complex, KDM6A promotes the activation of enhancers by MLL4/p300 and is responsible for controlling the synergy between H3K4me1 and H3K27ac [76,113]. Although KMT2D and KMT2C share some mutual activities, their mutations in PDAC arise independently [19,43,115,116]. Although mutations and decreased expression levels in both *KMT2D* and *KMT2C* are linked to a more favorable prognosis in PDAC in certain studies [43,82], one study shows the therapeutic advantages of targeting these methyltransferases in PDAC. This is particularly noteworthy in patients who exhibit higher levels of KMT2C/D expression. In other words, while these methyltransferases may typically indicate a better prognosis, targeting them could still hold therapeutic promise, especially in cases where their expression levels are higher [116].

KMT2A is a regulatory agent that regulates gene transcription by H3K4me3 activation on promoters [117,118]. In different types of cancers, KMT2A acts as an oncogenic factor and plays a significant role in hypoxia signaling, angiogenesis, and tumorigenesis [119]. KMT2A plays a multifaceted role in cancer, promoting tumor invasion and metastasis in colorectal cancer [120], driving oncogenesis in aggressive lymphoid and myeloid leukemias, and influencing cell viability, migration, and apoptosis in cervical cancer through modulation of apoptosis-related proteins [120,121]. Accordingly, *KMT2A* is potentially an oncogene, which underscores its importance in cellular function and malignancy. On the other hand, it has been demonstrated that the inactivation of KMT2A in *Men1*-deficient mice accelerated pancreatic tumorigenesis and contributed to poorer prognosis, with increased cell proliferation observed in the derived tumors [81]. Targeting the KMT2A protein subcomplex epigenetically diminished the tumorigenic characteristic of pancreatic adenosquamous carcinomas, implying the capability for KMT2A inhibition as an adjunct to chemotherapy and eventually enhancing clinical outcomes for pancreatic cancer patients [122].

Another tumor-promoting component of COMPASS-like complex genes affecting the SET and PHD domains is *KMT2B*, which is less studied [84]. KMT2B is known as a cell proliferation factor, and its inactivation results in defects in cell proliferation and apoptosis, suggesting the possible oncogenic role of KMT2B [123,124]. *KMT2B* has higher mutation rates in esophageal, gastric, cervical, and uterine carcinoma [77,125,126,127]. KMT2B may also contribute to modulating overall survival in other malignancies, including leukemia [128]. In addition to the oncogenic role of KMT2B in other cancers, pancreatic carcinoma displays an amplification in a zone that involves the *KMT2B* gene [129]. The oncogenic role of KMT2B in breast cancer cells is exerted by upregulating the expression of estrogen receptors [130]. KMT2B-mediated tumorigenesis in cervical cancer is conditioned by inducing angiogenesis and modulating the epithelial growth factor pathway, hinting that targeting epithelial growth factor receptors could boost therapeutic efficacy in cervical cancer patients exhibiting high KMT2B expression [127]. In terms of interacting with other components of the COMPASS-like complex, KMT2B, in collaboration with KDM6A, contributes to the orchestration of an oncogenic role in non-small cell lung carcinoma by modulating H3K4me3, indicating its potential as a prognostic marker and therapeutic target [131].

In summary, the COMPASS-like complex genes, including genes such as *KDM6A*, *KMT2D*, *KMT2C*, *KMT2A*, and *KMT2B*, are crucial to the epigenetic landscape of PDAC. As each gene contributes to PDAC pathogenesis through its unique functions and mechanisms, it presents opportunities for targeted therapy and advances our understanding of this challenging disease. Research on the role of these genes in PDAC provides a foundation for future translational studies and the development of new therapeutic approaches.

## 3. Uncovering Functions of COMPASS-like Complex Components to Unveil Therapeutic Vulnerability

As a result of recent advances in molecular profiling, our understanding of PDAC has been profoundly expanded. Specific subtypes of PDAC have been identified that have distinct therapeutic and prognostic implications. Initially, gene expression microarray analysis revealed three primary subtypes of PDACs: classical, quasi-mesenchymal, and exocrine-like [129]. According to Moffitt et al., further refinement using gene expression profiling with microdissection distinguished classical and basal-like PDACs [107]. RNA sequencing performed by the Australian Pancreatic Cancer Genome Initiative (APGI) revealed a more nuanced classification of PDACs into four different subtypes: squamous, pancreatic progenitor, immunogenic, and abnormally differentiated exocrine endocrine [132]. Complementing these findings, the Cancer Genome Atlas (TCGA) study, utilizing high tumor-content samples, identified two principal groups: classical/progenitor and basal-like/squamous [133,134,135]. These findings are in line with and support the previously mentioned subclassifications. The distinction between basal-like/squamous tumors has significant clinical importance, particularly since the basal-like/squamous group is associated with poorly differentiated tumors with a reduced overall survival rate [107,132,136,137].

Analysis of the genetic landscape of pancreatic cancer reveals a direct pattern of mutations, including mutations in *KRAS*, *TP53*, *CDKN2A*, and *SMAD4* [37,138]. However, comparing the genomes of primary tumors and the metastases from the same patients, there is a lack of diversity, and essential disease-causing mutations are uniform across all subclassifications [139]. Large-scale sequencing studies, such as whole exome and gene expression analyses, have greatly expanded our understanding of the molecular heterogeneity and complexity of PDAC beyond the well-known four major driver genes (*KRAS*, *TP53*, *SMAD4*, and *CDKN2A*), which are largely unactionable except the recent development of KRAS inhibitors. Furthermore, studies have identified recurrently altered pathways involving numerous mechanisms, including epigenetic dysregulation, that may contribute to pancreatic tumorigenesis [13,140]. Several studies found that cancers harboring mutations in the components of the COMPASS-like complex, including *KDM6A*, *KMT2C*, and *KMT2D*, were associated with poor differentiation and aggressive behavior [36,69]. Following these findings, Hissong et al. examined 103 primary and metastatic PDACs with comprehensive molecular profiling to determine specific clinicopathologic and molecular characteristics of mutations in COMPASS-like complex genes [35]. Consequently, they observed that PDACs with mutations in components of the COMPASS-like complex exhibited frequent poor differentiation areas with concurrent mutations in *TP53*, leading to decreased survival in the study group. Additionally, they examined two public data sets and concluded that PDACs with COMPASS-like complex alterations likely represent a subset of more aggressive PDACs with poor or squamous differentiation histologically and increased *TP53* mutation rates. These findings indicate that although genetic mutations may initiate cancer, its progression to metastatic stages may be heavily influenced by epigenetic mechanisms [35,73,141,142]. This understanding points to the need for a more comprehensive approach to treating PDACs, encompassing genetic and epigenetic factors to effectively combat this aggressive disease.

### 3.1. Super-Enhancer Regulation in Pancreatic Cancer: The Impact of KDM6A Mutations and BET Inhibitor Therapy

Based on next-generation sequencing data collected from various cohorts and human cell lines, studies have revealed the crucial role of super-enhancers (SEs) in the progression of pancreatic cancer, utilizing advanced methodologies including chromatin immunoprecipitation sequencing (ChIP-seq) and RNA sequencing (RNA-seq) across multiple human cell lines and tissue samples to analyze enhancer landscapes and their transcriptional outcomes [143,144,145]. It is becoming increasingly recognized that SEs contribute significantly to driving abnormal transcriptional programs that lead to cellular identity loss and metastasis [146,147,148]. Additionally, these studies have demonstrated the importance of the COMPASS-like complex components in regulating these SEs.

*KDM6A* is an essential gene that impacts developmental processes and cancer progression. The lack of KDM6A results in changes to the activity of primary enhancers and super-enhancers, which are crucial to controlling gene expression [36]. Consequently, this disruption in gene regulation during development can also facilitate cancer progression. Such disturbances in gene expression mainly affect the transcription of essential genes like *Np63*, *MYC*, and *RUNX3* [19,25,36,149,150,151]. These genes play critical roles in squamous cell differentiation and the spread of cancer, underscoring the importance of KDM6A in both developmental regulation and cancer prevention (Figure 1). According to the findings, loss of KDM6A activates gene signatures associated with IL-6 signaling, a pathway implicated in desmoplasia [152,153]. In addition, the upregulation of RUNX3 may contribute to the formation of desmoplastic and metastatic niches [151]. SE reprogramming is a pivotal step in metastatic pancreatic cancer across multiple subtypes, including those of squamous-like and pancreatic progenitor subtypes [142]. This understanding opens avenues for utilizing SE reprogramming signatures as biomarkers to stratify patients with aggressive forms of PDAC, potentially guiding more personalized treatment approaches. Fortunately, it has been demonstrated that despite varying SE reprogramming patterns, these subtypes exhibit sensitivity to BET inhibitors. Several BET inhibitors have shown promise as adjuvant therapies in treating squamous-like and metastatic pancreatic cancers, particularly in KDM6A-deficient PDACs [36]. The identification of sensitivity to BET inhibitors in specific PDAC subtypes could lead to targeted therapy protocols, improving patient outcomes by selectively inhibiting cancer progression where KDM6A mutations are present. They enhance the effectiveness of current therapies by restoring cell identity. The BET inhibitor JQ1 is particularly effective in inhibiting squamous differentiation and controlling *Kdm6a*-null pancreatic cancer in vivo. Further research could explore the integration of JQ1 and similar inhibitors into standard treatment regimens, potentially establishing a new standard of care for this patient subgroup. Hence, BET inhibitors like JQ1 may be useful for reversing the effects of *KDM6A* mutations in specific pancreatic cancer types.

### 3.2. Activin A and the Noncanonical p38 MAPK Pathway: Exploring the Molecular Mechanisms of PDAC Progression

Activin A, a member of the TGF-β superfamily, is implicated in PDAC progression [154]. Elevated blood levels of activin A in PDAC patients correlate with shorter survival times and increased tumor growth. This correlation suggests the potential for using activin A as a biomarker and candidate therapeutic target for patient prognosis and therapy customization. Yi Z et al. discovered that the loss of the *KDM6A* gene contributes to the development of an aggressive, undifferentiated subtype of PDAC [69]. This process is primarily mediated by a noncanonical activin A pathway involving p38 MAPK (Figure 2). While normally regulating cell differentiation, the absence of functional KDM6A promotes the mesenchymal identity of cancer cells, facilitating tumor growth and metastasis, thereby exacerbating PDAC aggressiveness. Understanding this mechanism could enable the identification of patients at risk for aggressive PDAC and the development of targeted therapies. In another investigation by S Lu et al., the role of KMT2D in pancreatic cancer was examined [108]. Utilizing small interfering RNA to knock down KMT2D expression in pancreatic cancer cell lines, the researchers assessed changes in miR-147b expression and activin A levels using RT-qPCR and ELISA, respectively. TGF-β-induced upregulation of miR-147b levels results in KMT2D mRNA degradation or translation inhibition. Diminished KMT2D levels lead to increased production and release of activin A, which activates the noncanonical p38 MAPK-dependent pathway and drives the epithelial–mesenchymal transition in cancer cells to promote invasiveness (Figure 2).

These studies underscore the pivotal roles of KDM6A and KMT2D in PDAC progression, particularly through the noncanonical activin A pathway. They offer insights into potential new therapeutic targets for combating this challenging cancer subtype. Expanding these insights could facilitate the development of diagnostic and therapeutic strategies that are personalized to the molecular profile of the patient’s tumor, potentially improving outcomes in PDAC treatment.

### 3.3. The Role of Histone Modifications and Therapeutic Implications

The intricate landscape of histone modifications, including methylation and acetylation, is crucial in determining chromatin structure and gene expression, leading to various cellular outcomes. Central to this regulatory framework is the polycomb repressor complex-2 (PRC2) and its key component enhancer of zeste homolog-2 (EZH2), which orchestrates the deposition of H3K27me3, a hallmark of gene silencing [155,156,157]. This action contrasts gene activation signals, such as trimethylation at histone 3 lysine 4 (H3K4), highlighting the complex dynamics governing gene expression patterns, particularly in PDAC. Utilizing ChIP-seq to map these histone modifications across different PDAC cell lines, we can provide a high-resolution view of the epigenetic alterations associated with tumor aggressiveness. This molecular insight can facilitate the identification of aggressive PDAC cases through biomarker-driven patient stratification, where high levels of H3K27me3 could predict a poor prognosis and greater need for targeted therapeutic intervention.

Recent research underscores the significance of KDM6A, an antagonist to PRC2, in the PDAC context, suggesting that tumors lacking KDM6A activity or expression may be uniquely sensitive to EZH2 inhibition strategies [67]. Andricovich et al. employed a combination of gene silencing techniques and pharmacological inhibition to evaluate the therapeutic efficacy of EZH2 inhibitors in PDAC models, both in vitro and in vivo. Studies, including that by Andricovich et al., demonstrate the efficacy of EZH2 inhibitors in both KDM6A-deficient and -proficient PDAC cells, hinting at KDM6A’s broader role in tumor suppression, stem cell maintenance, and developmental processes [36]. This finding enriches our understanding of KDM6A beyond its demethylase function, suggesting its involvement in a wider regulatory network. Strategies for clinical application include using KDM6A expression levels to guide the use of EZH2 inhibitors as personalized treatments, potentially improving outcomes for patients with specific genetic profiles.

Further complexity is added by the role of histone deacetylases (HDACs) in repressing gene transcription. Watanabe et al. used CRISPR-Cas9 to specifically knockout the *KDM6A* gene in their research. They then conducted assays to observe how this knockout affected acetylation levels and transcriptional activity at the promoters of important genes. Their findings showed that the absence of KDM6A resulted in decreased acetylation of H3K27 at these promoters, including that of the tumor suppressor gene *CDKN1A*. This reduction was attributed to hindered recruitment of the histone acetyltransferase p300 [70]. This insight reinforces the potential of HDAC inhibitors as targeted therapies for KDM6A-deficient PDAC cells, aiming to restore the expression of crucial tumor suppressors and counteract PDAC progression. Leveraging this mechanism, diagnostic tools could be developed to detect KDM6A and H3K27 acetylation statuses, optimizing HDAC inhibitor therapy application.

In conclusion, the exploration of histone modifications, mainly through the lens of KDM6A deficiency in PDAC, reveals a complex yet promising field of potential therapeutic interventions. By inhibiting key enzymes like EZH2 and HDACs, new avenues for treating PDAC are unveiled, emphasizing the need for a deepened understanding of epigenetic mechanisms. This comprehensive approach highlights the significance of epigenetic regulation in cancer biology and the promising horizon for developing effective PDAC treatments. These insights offer a foundation for translational research aimed at developing precise, epigenetically informed therapeutic strategies for PDAC.

### 3.4. Altering the Tumor Microenvironment and Promoting Cancer Progression

The impact of KDM6A deficiency on the tumor microenvironment (TME) in PDAC has been recently described [158]. The absence of KDM6A leads to an immunosuppressive TME. This is characterized by increased tumor-associated neutrophils (TANs) that promote tumor growth. Additionally, a marked reduction in CD8+ T cells and B cells indicates a compromised immune response to the tumor. This deficiency is closely linked to accelerated cancer progression [158].

CXCL1 functions as a chemoattractant, guiding the movement of immune cells through concentration gradients. CXCL1 plays a dual role by facilitating the recruitment of TANs, thereby promoting tumor growth and diminishing the CD8+ T cell population, weakening the immune defense against the tumor [159,160]. The study by Yang et al. reinforced these findings, emphasizing the impact of KDM6A deficiency on PDAC’s TME [158]. The study also highlights the potential of targeting the CXCLs–CXCR2 axis as a therapeutic strategy in PDAC cases with KDM6A loss. They used a genetically engineered, pancreas-specific *KDM6A*-knockout PDAC mouse model and human PDAC tissue samples to demonstrate that KDM6A loss correlates with increased TANs and neutrophil extracellular trap (NET) formation, which are known to contribute to PDAC progression (Figure 3). Moreover, transcriptomic analysis showed that the expression of many chemotactic cytokines, especially CXCL1, were upregulated in *KDM6A*-knockout PDAC cells. This molecular insight suggests that targeting the CXCLs–CXCR2 axis could serve as a therapeutic strategy in PDAC cases with KDM6A loss. Additionally, identifying patients with altered KDM6A expression could facilitate the stratification of PDAC cases into those likely to respond to CXCL1-neutralizing treatments. In a mouse model, treatment with a CXCL1-neutralizing antibody blocked the chemotactic and NET-promoting properties of KDM6A-deficient PDAC cells and suppressed tumor growth, highlighting its potential as a targeted treatment strategy. This approach could be further validated in clinical trials aimed at improving patient outcomes by exploiting this newly understood pathway.

### 3.5. Immune Checkpoint Therapy

Immune checkpoint blockade (ICB) has demonstrated remarkable clinical efficacy across several cancer types; however, only a subset of PDAC patients respond to this treatment. Wang et al. conducted a study revealing that *KMT2D* mutations sensitize various cancer types to immune checkpoint therapy by reshaping the tumor microenvironment [161]. They employed pooled mutagenic screening within CRISPR-mediated genetically engineered mouse models specifically designed for ICB settings, which enabled the identification of KMT2D as a pivotal modulator of ICB response across multiple cancers. This methodological approach allows for the precise manipulation and observation of genetic alterations within a controlled environment, ensuring the robustness of the findings.

The loss of KMT2D function resulted in increased DNA damage and mutation burden, chromatin remodeling, intron retention, and activation of transposable elements. Furthermore, *KMT2D*-mutant cells exhibited heightened protein turnover and IFNγ-stimulated antigen presentation. Consequently, *KMT2D*-mutant tumors in both mice and humans displayed enhanced immune infiltration, characterized by elevated levels of cytokines like CCL2, CCL5, CCL22, and CXCL9 that attract antigen-presenting cells and CD8+ T cells into the tumor microenvironment. These molecular insights suggest that patients with PDAC harboring *KMT2D* mutations could be stratified for more aggressive and tailored immunotherapeutic approaches. Further research could explore the utility of combining traditional therapies with ICB to potentially improve outcomes for these genetically distinct groups of PDAC patients.

These immune cell infiltrates were further accentuated following treatment with programmed cell death protein 1 (PD-1) inhibitors, suggesting that *KMT2D* mutations could identify pancreatic cancer patients who may benefit from ICB therapy.

Subsequently, Li et al. confirmed this hypothesis in a case study where *KMT2D* mutations were identified in pancreatic cancer patients who exhibited remarkable responses to immunotherapy, particularly ICBs [162]. They utilized genomic sequencing and immune profiling to demonstrate how these mutations alter the tumor microenvironment, thereby facilitating increased infiltration and activation of effector immune cells, particularly CD8+ T cells. This evidence supports developing targeted treatment protocols that specifically address the unique genetic and immunological profiles of PDAC patients, enhancing precision medicine strategies.

Expanding on this, Huang et al. explored the role of *KMT2C* mutations in PDAC [163]. Their research emphasized tumor mutation burden (TMB) as a predictive biomarker for immunotherapy response, particularly with ICBs targeting PD-L1 and CTLA-4 proteins [164,165]. Through extensive genomic and proteomic analyses, they identified correlations between *KMT2C* mutations and changes in metabolic pathways and immune cell composition, reinforcing the potential of these mutations as prognostic biomarkers and guides for tailored immunotherapy in PDAC treatment. This further highlights the potential for clinical application using *KMT2C* mutations to not only predict but also potentially drive the therapeutic strategy in a clinical setting. While ICBs have shown efficacy in various cancers, including PDAC, reliance solely on PD-L1 expression for predicting immunotherapy response is limited. *KMT2C* mutations, associated with higher TMB and poorer prognosis in PDAC patients, influence metabolic and protein-related pathways as well as the composition of immune cells. This positions *KMT2C* mutations as potential prognostic biomarkers and guides for tailored immunotherapy in PDAC treatment, underscoring their significance in PDAC’s molecular landscape and therapeutic strategies.

### 3.6. Metabolic and Protein-Related Pathways

The study by Wang et al. has uncovered a critical function of KMT2A in protein glycosylation and endoplasmic reticulum (ER) functionality in PDAC [166]. Employing transcriptomic analysis and proteomic profiling, they found that KMT2A deficiency leads to an increased activation of the unfolded protein response (UPR) and cell apoptosis, highlighting its essential role in maintaining protein homeostasis. Additionally, KMT2A regulates genes involved in glycometabolism and protein glycosylation, indicating potential therapeutic targets in PDAC treatment. The team utilized RNA sequencing to assess gene expression changes and employed mass spectrometry to analyze protein glycosylation patterns, further substantiating the molecular pathways impacted by KMT2A.

To translate these molecular insights into clinical practice, the identification of KMT2A-linked biomarkers could enhance the stratification of patients with aggressive PDAC, potentially guiding the development of personalized treatment strategies. By influencing glycometabolism, KMT2A affects the energy supply and biosynthetic precursors necessary for rapid cancer cell proliferation. Meanwhile, alterations in protein glycosylation patterns can impact cell–cell communication, adhesion, and immune recognition, further facilitating the evasion of cancerous cells from the immune system and their spread to other parts of the body. Understanding the pivotal role of KMT2A in these processes highlights its potential as a therapeutic target. Targeting KMT2A or the pathways it regulates could disrupt the metabolic flexibility and glycosylation-dependent mechanisms that PDAC cells rely on, offering a novel approach to therapy. Such strategies might involve the development of small molecule inhibitors that specifically inhibit KMT2A activity or drugs that modulate the expression of genes in the glycometabolism and glycosylation pathways under KMT2A’s control. Through targeted intervention at the molecular level, it may be possible to significantly impair PDAC cell survival and proliferation, paving the way for more effective treatments for this devastating disease. Complementing these findings, Li et al. delved into the impact of KMT2D expression on the efficacy of L48H37, a curcumin analog, in PDAC cells [167]. They observed that L48H37 induces apoptosis through the ER stress-associated signaling pathway, with its anti-cancer effects enhanced by reduced KMT2D levels. This suggests that treatments like L48H37 could be promising in treating PDACs with mutations in the COMPASS-like complex gene families, including *KMT2A* and *KMT2D*. Table 2 summarizes potential therapeutic targets aiming at COMPASS-like complex gene pathways in PDAC.

## 4. Conclusions

The intricate interplay between genetic mutations and epigenetic dysregulation in PDAC underscores the complexity of this malignancy. By elucidating the role of the COMPASS-like complex in PDAC pathogenesis, promising avenues for therapeutic intervention can be identified. Targeting key components of this complex, such as KDM6A and members of the KMT2 family, can disrupt oncogenic signaling pathways and restore normal cellular function. Additionally, insights into the tumor microenvironment and immune checkpoint regulation provide further opportunities for personalized treatment strategies. As we unravel the molecular intricacies of PDAC, collaboration between researchers and clinicians will be essential to translate these findings into clinical practice and improve patient outcomes.

## Figures and Tables

**Figure 1 ijms-25-05069-f001:**
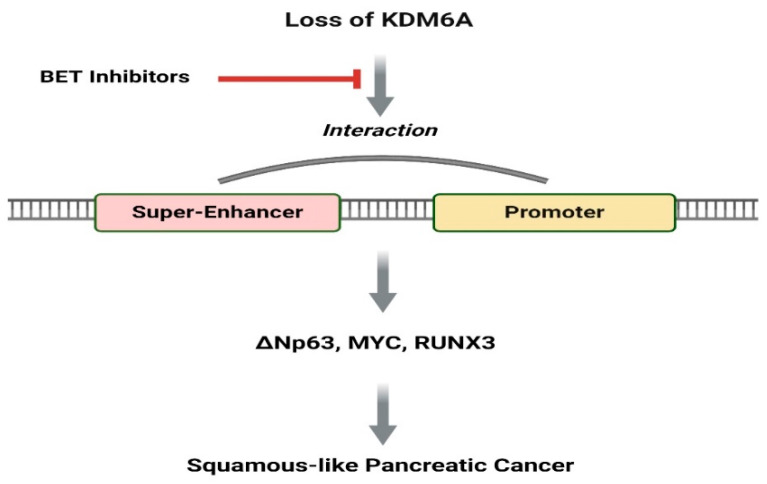
KDM6A loss leads to the activation of super-enhancers (SEs) that regulate oncogenes such as *ΔNp63*, *MYC*, and *RUNX3*, contributing to cancer development. Treatment with JQ1, a drug that disrupts long-range SE–promoter interactions, has been shown to help restore pancreatic cancer cell identity.

**Figure 2 ijms-25-05069-f002:**
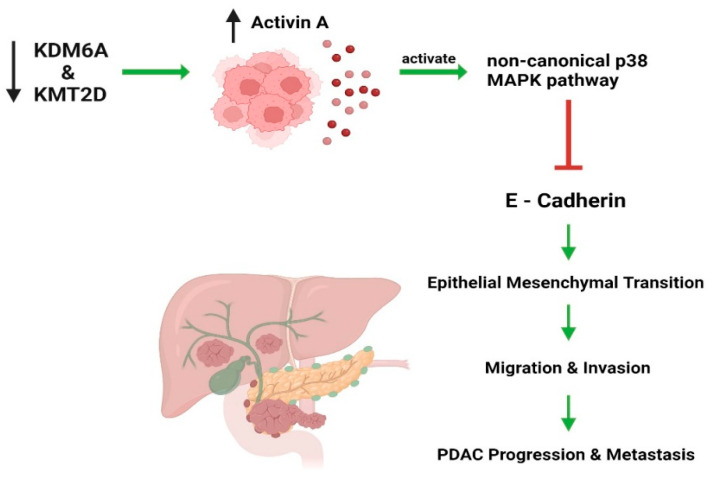
The reduction in KDM6A and KMT2D expression results in heightened synthesis and secretion of activin A. This increase in activin A triggers the activation of the noncanonical p38 MAPK pathway, consequently modifying the behavior of cancer cells to enhance their invasiveness.

**Figure 3 ijms-25-05069-f003:**
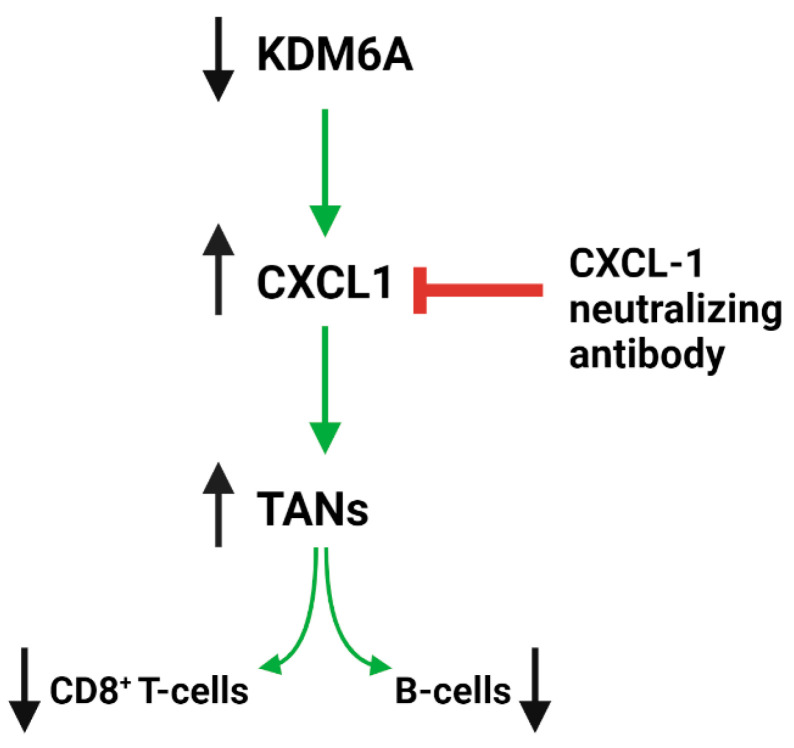
The impact of KDM6A deficiency on the TME in PDAC. The absence of KDM6A leads to a notable shift in the TME towards an immunosuppressive state. Key features of this altered environment include an increased presence of tumor-associated neutrophils (TANs), which promotes tumor growth. Additionally, there is a substantial decrease in immune-effective cells such as CD8+ T cells and B cells, indicating a compromised immune response to the tumor. Central to this process is the role of CXCL1 that facilitates the recruitment of TANs and subsequently leads to a reduction in CD8+ T cell populations, further weakening the immune system’s defense against the tumor.

**Table 1 ijms-25-05069-t001:** Summary of roles of main COMPASS-like complex genes in PDAC.

Gene	Main Function	Effects on PDAC Pathology	Interaction with Other CLC * Genes
** *KDM6A* **	Histone demethylation of H3K27	Tumor suppressive role; loss of function mutations contribute to dedifferentiation, enhanced cancer cell migration, and invasion, and poor prognosis	Interacts with KMT2D and KMT2C in the COMPASS-like complex
** *KMT2D* **	Histone methylation of H3K4	Tumor suppressive role; loss of function mutations lead to increased cancer cell invasion and migration; and correlates with poor prognosis	Interacts with KDM6A and KMT2C in the COMPASS-like complex
** *KMT2C* **	Histone methylation of H3K4	Potential tumor suppressor; involved in enhancer activity regulation; mutations linked to better prognosis in some studies	Interacts with KDM6A and KMT2D in the COMPASS-like complex
** *KMT2A* **	Histone methylation of H3K4	Oncogenic role; promotes tumor invasion, metastasis, tumorigenesis	Interacts with KMT2B in the COMPASS-like complex
** *KMT2B* **	Histone methylation of H3K4	Oncogenic role; contributes to cell proliferation; mutations associated with poorer prognosis in some cancers	Interacts with KMT2A in the COMPASS-like complex

* CLC: COMPASS-like complex.

**Table 2 ijms-25-05069-t002:** Summary of potential therapeutic targets aiming at COMPASS-like complex gene pathways in PDAC.

Therapeutic Target	Drug Name	References/NCT Numbers	Stage of Development	Cancer Type	Results
**HDAC inhibitors**	Entinostat	NCT05053971	Phase I/II	Solid tumors, PDAC and lymphomas	Not completed.
	Entinostat	NCT03925428	Phase I	Solid tumors	No results published yet.
	Vorinostat	Watanabe, S., et al. [70]	Preclinical	PDAC	KDM6A knockout cells exhibit increased sensitivity to HDAC inhibitors compared to KDM6A wild-type cells.
	Romidepsin	NCT04257448	Phase I/II	PDAC	No results published yet.
**BET inhibitors**	JQ1	Andricovich, J., et al. [36]	Preclinical	PDAC	KDM6A-deficient pancreatic cancer is selectively sensitive to BET inhibitors.
**CXCL1-neutralizing antibodies**	N/A	Yang, J., et al. [158]	Preclinical	PDAC	CXCL1-neutralizing antibodies effectively suppressed KDM6A-deficient PDAC growth by blocking the chemotactic and NET-promoting properties of KDM6A-deficient PDAC cells.
**Immune checkpoint inhibitors**	Anti-PD-L1	Wang, G., et al. [161]	Preclinical	Multiple Cancers	*Kmt2d* deficiency increased sensitivity to immune checkpoint blockade (ICB).
	Durvalumab	NCT04257448	Phase I/II	PDAC	No results published yet

## Data Availability

Not applicable.

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
