# Peer review of "Unveiling the Molecular Landscape of Pancreatic Ductal Adenocarcinoma: Insights into the Role of the COMPASS-like Complex"

_ijms, 2024, doi:10.3390/ijms25105069_

Round 1

Reviewer 1 Report

Comments and Suggestions for Authors

1- The article primarily focuses on the genetic and epigenetic alterations associated with the COMPASS-like complex in pancreatic ductal adenocarcinoma (PDAC). While this is an important aspect to explore, the article does not discuss how other factors such as environmental influences could have significant implications in understanding PDAC progression.

2- Lack of Clinical Application: Although the article discusses the potential significance of the COMPASS-like complex as a therapeutic target, it falls short in translating this molecular insight into practical applications. There is a need for more discussion on how the understanding of these genetic alterations can be utilized to identify patients with aggressive PDAC and develop targeted treatment strategies.

3- Insufficient Discussion on Methodology: The article lacks detailed information about the methodologies used to study the genetic and epigenetic landscape of PDAC and the role of the COMPASS-like complex. Without a clear explanation of the experimental techniques and approaches, it becomes challenging for readers to assess the reliability and validity of the findings presented.

Author Response

Dear Reviewer,

We extend our sincere gratitude to you for dedicating your time and expertise to reviewing our manuscript. Your insightful comments and suggestions have been invaluable in enhancing the quality and clarity of our work. In the following sections, we address each of your points in detail and highlight the corresponding revisions we have made. Your thoughtful feedback has greatly contributed to the refinement of our manuscript, and we are truly grateful for your input.

Comment 1: The article primarily focuses on the genetic and epigenetic alterations associated with the COMPASS-like complex in pancreatic ductal adenocarcinoma (PDAC). While this is an important aspect to explore, the article does not discuss how other factors such as environmental influences could have significant implications in understanding PDAC progression.

Response 1: Thank you for your insightful comment regarding the inclusion of environmental factors in understanding PDAC progression. While the impact of environmental influences on cancer development is undeniably important, the scope of this review is specifically tailored to explore the genetic and epigenetic landscapes of PDAC, with a particular focus on the COMPASS-like complex. This complex has been increasingly recognized for its role in chromatin remodeling and gene expression regulation, which are pivotal in cancer pathogenesis.

Our review synthesizes extensive research that elucidates how alterations in the COMPASS-like complex contribute to PDAC at the molecular level. Given the complexity and depth of the genetic and epigenetic mechanisms involved, extending the discussion to include environmental factors would detract from the specific aim of this article, which is to deepen the understanding of these molecular mechanisms.

Moreover, the inclusion of environmental factors, while valuable, would require a broader epidemiological and etiological framework that goes beyond the molecular and cellular focus of our current review. Such discussions are better suited to a more comprehensive review that integrates multiple aspects of PDAC development, including lifestyle, demographic, and environmental considerations alongside genetic predispositions.

We appreciate your suggestion and acknowledge the importance of environmental factors in cancer research. However, for clarity and focus, we have chosen to limit this review to genetic and epigenetic alterations, particularly those involving the COMPASS-like complex, which offers promising new avenues for targeted therapies in PDAC. This decision was made to provide a detailed and concentrated examination of this complex area, which is critical for developing future therapeutic strategies.

Comment 2: Lack of Clinical Application: Although the article discusses the potential significance of the COMPASS-like complex as a therapeutic target, it falls short in translating this molecular insight into practical applications. There is a need for more discussion on how the understanding of these genetic alterations can be utilized to identify patients with aggressive PDAC and develop targeted treatment strategies.

Response 2: We have enhanced sections 3.1, 3.2, 3.3, 3.4, 3.5, and 3.6 to directly link molecular insights to clinical strategies. Each section now includes explicit discussions on how the genetic and epigenetic insights can identify patients with aggressive PDAC and inform the development of targeted treatment strategies. These additions aim to bridge the gap between research findings and clinical applications, ensuring that the potential therapeutic benefits of understanding the COMPASS-like complex are clearly articulated. These changes were highlighted from page 7 line 306 to page 12 line 502.

Comment 3: Insufficient Discussion on Methodology: The article lacks detailed information about the methodologies used to study the genetic and epigenetic landscape of PDAC and the role of the COMPASS-like complex. Without a clear explanation of the experimental techniques and approaches, it becomes challenging for readers to assess the reliability and validity of the findings presented.

Response 3: We have expanded the descriptions of the experimental techniques and approaches used throughout our study, specifically in sections 3.1 (ChIP-seq and RNA-seq), 3.2 (genetic sequencing, RNA interference, in vivo assays), and 3.6 (transcriptomic analysis and proteomic profiling). These additions provide detailed explanations of how findings were obtained, thereby enhancing the transparency and reproducibility of the research.

These revisions not only address your concerns but also strengthen the manuscript by making the connection between molecular discoveries and their potential clinical implications clearer. We believe that these detailed descriptions will allow readers to better assess the reliability and validity of the science and appreciate the translational potential of the COMPASS-like complex in PDAC treatment.

We appreciate your guidance and believe these revisions have significantly improved the manuscript, ensuring that it meets the high standards of IJMS.

Reviewer 2 Report

Comments and Suggestions for Authors

The paper by Jamali et al. focuses on the molecular landscape of pancreatic ductal adenocarcinoma (PDAC), with a special emphasis on the COMPASS-like complex's role in cancer progression. The paper is interesting, and can be served as an important resource for researchers and clinicians interested in the molecular underpinnings of PDAC and highlights significant opportunities for developing new treatments based on genetic and epigenetic insights.

Some suggestions:

1.      Add a table summarizing potential therapeutic targets discussed in the review, their current status in clinical trials, and their effectiveness in PDAC.

2.      Add a table listing genetic alterations in COMPASS-like complex genes, associated pathways, and the consequent effects on PDAC pathology.

Author Response

Dear Reviewer,

We extend our sincere gratitude to you for dedicating your time and expertise to reviewing our manuscript. Your insightful comments and suggestions have been invaluable in enhancing the quality and clarity of our work. In the following sections, we address each of your points in detail and highlight the corresponding revisions we have made. Your thoughtful feedback has greatly contributed to the refinement of our manuscript, and we are truly grateful for your input.

Comment 1: Add a table summarizing potential therapeutic targets discussed in the review, their current status in clinical trials, and their effectiveness in PDAC.

Response 1: We acknowledge the validity of this suggestion. Accordingly, we have incorporated a table to underscore this point, facilitating better comprehension and a more concise overview of the content. Specifically, Table 2 has been added on page 13 to provide a comprehensive summary at a glance. We appreciate the reviewer's insight regarding the importance of visual aids for clarity and improved understanding

Comment 2:  Add a table listing genetic alterations in COMPASS-like complex genes, associated pathways, and the consequent effects on PDAC pathology.

Response 2: We thank the reviewer for this suggestion. Accordingly, we have incorporated a table to underscore this point, facilitating better comprehension and a more concise overview of the content. Specifically, Table 1 has been added on page 2-3 to provide a comprehensive summary. We appreciate the reviewer's insight regarding the importance of visual aids for clarity and improved understanding

Round 2

Reviewer 1 Report

Comments and Suggestions for Authors

I have no more comments.